# Recent Advances in Biomimetics for the Development of Bio-Inspired Prosthetic Limbs

**DOI:** 10.3390/biomimetics9050273

**Published:** 2024-04-30

**Authors:** Pavitra Varaganti, Soonmin Seo

**Affiliations:** Department of Bionano Technology, Gachon University, Seongnam 13120, Republic of Korea; pavitra11@gachon.ac.kr

**Keywords:** biomimetics, prosthetics, bio-inspired, trends, integration

## Abstract

Recent advancements in biomimetics have spurred significant innovations in prosthetic limb development by leveraging the intricate designs and mechanisms found in nature. Biomimetics, also known as “nature-inspired engineering”, involves studying and emulating biological systems to address complex human challenges. This comprehensive review provides insights into the latest trends in biomimetic prosthetics, focusing on leveraging knowledge from natural biomechanics, sensory feedback mechanisms, and control systems to closely mimic biological appendages. Highlighted breakthroughs include the integration of cutting-edge materials and manufacturing techniques such as 3D printing, facilitating seamless anatomical integration of prosthetic limbs. Additionally, the incorporation of neural interfaces and sensory feedback systems enhances control and movement, while technologies like 3D scanning enable personalized customization, optimizing comfort and functionality for individual users. Ongoing research efforts in biomimetics hold promise for further advancements, offering enhanced mobility and integration for individuals with limb loss or impairment. This review illuminates the dynamic landscape of biomimetic prosthetic technology, emphasizing its transformative potential in rehabilitation and assistive technologies. It envisions a future where prosthetic solutions seamlessly integrate with the human body, augmenting both mobility and quality of life.

## 1. Introduction

Biomimetics, the emulation of nature’s designs and processes to address human challenges, is a rapidly evolving field that has garnered significant attention, particularly within the realm of prosthetic limb development [1,2]. It encompasses a multidisciplinary approach, drawing insights from biology, materials science, robotics, and bioengineering. By tapping into the efficiency and effectiveness of natural systems, biomimetic prosthetics offer promising advancements in functionality, comfort, and integration with the human body [3]. This field seeks to harness the solutions found in nature to tackle complex human challenges across various domains, including medicine, architecture, and technology. Specifically, in the context of prosthetic limb development, biomimetics involves a comprehensive study of biomechanics, structural composition, and sensory feedback mechanisms of natural limbs to inform the design and engineering of artificial alternatives [4,5]. Through the replication of biological structures and materials, these prosthetic limbs aim to closely mimic the natural movements and sensations of real limbs, thereby enhancing the overall quality of life for individuals with limb loss [6]. Moreover, biomimetic designs hold significant potential for reducing the risk of rejection and minimizing environmental impact, positioning them as a crucial avenue for improving the lives of amputees globally [7].

Recent advancements in biomimetic prosthetics have been facilitated by the convergence of various scientific disciplines [8,9,10]. Materials science has played a pivotal role in enabling the development of advanced materials that closely mimic the properties of biological tissues, thereby enhancing the comfort and functionality of prosthetic limbs [11]. Robotics has contributed significantly to the creation of prosthetic devices with sophisticated control systems, allowing users to execute a wide range of movements with precision and ease [12]. Additionally, developments in neural interfaces and sensor technologies have empowered prosthetic limbs to provide sensory feedback to users, thereby improving their sense of embodiment and interaction with the environment [6,13]. The significance of biomimetic prosthetics in the present context cannot be overstated. With the increasing prevalence of limb loss worldwide, driven by factors such as traumatic injuries, congenital conditions, and surgical interventions, there is a growing demand for advanced prosthetic solutions [14]. While traditional prosthetic devices offer basic functionality, they often fall short in terms of comfort, usability, and natural feel. Biomimetic prosthetics offer a paradigm shift by aiming to replicate the intricate functions and aesthetics of natural limbs, thereby enhancing the overall quality of life for individuals with limb loss [5,15]. One of the key advantages of biomimetic prosthetics is their ability to provide a more natural and intuitive user experience [16]. By closely mimicking the biomechanics and sensory feedback mechanisms of biological limbs, these prosthetic devices afford users greater control and proprioception, enabling more fluid and coordinated movements [5,15]. This not only enhances the user’s ability to perform daily tasks but also fosters a sense of confidence and connection to their body, ultimately contributing to improved psychological well-being.

Moreover, biomimetic prosthetics prioritize biocompatibility and sustainability, addressing key challenges associated with traditional prosthetic devices [17]. By leveraging materials and technologies inspired by nature, these prosthetic limbs aim to reduce the risk of rejection and adverse reactions, thereby improving long-term outcomes for users [6,18]. Additionally, biomimetic design principles promote sustainability by minimizing environmental impact and resource consumption, aligning with global efforts towards more eco-friendly solutions [1,19]. Recent research in biomimetic prosthetics has concentrated on advancing the state-of-the-art in materials science, robotics, neural interfaces, and sensor technologies [20,21]. One notable area of innovation is the development of biomimetic materials that closely resemble the properties of biological tissues. These materials, such as shape-memory alloys and hydrogels, offer improved flexibility, durability, and biocompatibility compared to traditional prosthetic materials, thereby enhancing the comfort and performance of prosthetic limbs [22]. Advancements in robotics have led to the creation of prosthetic devices with sophisticated control systems, enabling users to execute complex movements with precision and accuracy [12]. By integrating artificial intelligence and machine learning algorithms, these prosthetic limbs can adapt to the user’s preferences and movements, providing a more personalized and intuitive user experience [23]. Additionally, developments in neural interfaces have enabled direct communication between prosthetic devices and the user’s nervous system, allowing for seamless integration and a natural control of the limb [24]. Sensor technologies play a crucial role in biomimetic prosthetics by providing real-time feedback to users and clinicians [25]. By incorporating sensors that detect pressure, temperature, and motion, prosthetic limbs can offer users sensory feedback, enhancing their perception of touch and proprioception [26]. This not only improves the user’s ability to interact with their environment but also reduces the risk of injury and enhances overall safety.

Biomimetic prosthetics have demonstrated promising results in various real-world applications, showcasing their transformative potential in rehabilitation and assistive technologies [27]. Case studies have highlighted the benefits of biomimetic design principles in improving the functionality, comfort, and aesthetics of prosthetic limbs [28,29]. For example, prosthetic limbs inspired by the biomechanics of natural limbs have been shown to provide users with greater mobility and dexterity, enabling them to perform activities of daily living with ease [5,6]. Moreover, biomimetic prosthetics have been instrumental in enhancing the quality of life for individuals with limb loss in diverse populations [15]. From athletes seeking to regain peak performance to elderly individuals aiming to maintain independence, biomimetic prosthetic limbs offer tailored solutions that meet the unique needs and preferences of users [30]. Additionally, biomimetic design principles have been applied in the development of prosthetic devices for specialized applications, such as prosthetic hands with advanced grip patterns for fine motor control [31]. Through continued research and innovation, biomimetic prosthetics have the potential to significantly improve the lives of individuals with limb loss worldwide, providing them with enhanced mobility, independence, and quality of life. The aim of this article is to provide a comprehensive overview of the latest trends and innovations driving the development of biomimetic prosthetics. By examining recent breakthroughs in materials science, robotics, neural interfaces, and sensor technologies, the review aims to elucidate the transformative potential of biomimetic design principles in prosthetic limb development. Moreover, the article seeks to showcase compelling case studies and real-world applications where biomimetic prosthetics have demonstrably improved the lives of individuals with limb loss, thereby highlighting the importance of this field in rehabilitation and assistive technologies. Through a systematic analysis of these advancements, the review aims to provide valuable insights into the evolving landscape of biomimetic prosthetics and guide future research and clinical implementation efforts.

## 2. Biomimetic Design Strategies

Biomimetic design strategies in prosthetic limb development encompass a comprehensive approach rooted in the understanding and replication of biological systems [31]. By delving into the intricate interaction of musculoskeletal components, sensory feedback mechanisms, and control systems in living organisms, researchers aim to fabricate prosthetic devices that closely emulate natural limbs [32]. Figure 1 illustrates various biomimetic approaches utilized in the development of bio-inspired prosthetic limbs.

Bionic limbs featuring muscle-like actuators replicate the functionality of natural muscles, enabling prosthetic limbs to achieve lifelike movement [27]. Sensory feedback systems are integrated to furnish users with touch and proprioceptive feedback, augmenting their perception and control over the prosthetic limb [33]. Biomimetic design principles are harnessed to mimic the structure and function of natural limbs, resulting in prosthetic limbs with enhanced aesthetics and functionality [34]. Bio-inspired control strategies entail the implementation of algorithms grounded in biological neural systems, facilitating intuitive movement of the prosthetic limb. Energy harvesting and storage mechanisms, inspired by nature, are incorporated to bolster the autonomy of prosthetic limbs by efficiently harvesting and storing energy [35]. Adaptive materials and interfaces adapt to the user’s requirements and environmental changes, thereby improving comfort and usability. Tendon-driven actuation systems transmit motion in a manner akin to natural limb movement, contributing to the lifelike functionality of the prosthetic limb [36]. Lastly, bio-inspired skin and texture are employed to fabricate prosthetic surfaces that closely resemble biological skin, providing users with a more natural appearance and tactile experience [37]. These aspects of biomimetic prosthetic design are underpinned by two core principles: understanding biological systems and leveraging advanced materials and manufacturing techniques. Through meticulous analysis and innovative fabrication methods, biomimetic prosthetics aim to enhance user experience and functionality, bridging the gap between artificial devices and natural appendages.

In prosthetic limb development, biomimetic design is pivotal, aiming to replicate biological intricacies. By studying musculoskeletal dynamics, sensory feedback, and control mechanisms in organisms, researchers create prosthetics that mimic natural limbs. These include muscle-like actuators for lifelike movement and sensory feedback systems for enhanced control. Biomimetic principles drive aesthetic and functional improvements, integrating bio-inspired materials, energy harvesting, and tendon-driven actuation. Deep biological understanding and advanced manufacturing advance this field, bridging the gap between artificial and natural limbs, enhancing the lives of amputees.

### 2.1. Understanding Biological Systems

Building upon the principles of biomimetic design strategies outlined in the previous section, a fundamental aspect lies in comprehensively understanding biological systems. This encompasses the intricate interplay of musculoskeletal components, sensory feedback mechanisms, and control systems within living organisms [38]. Such holistic comprehension is paramount for developing prosthetic devices that closely mimic the structure and functionality of natural limbs [39]. Central to this endeavor is the exploration of the musculoskeletal system, which governs movement and stability in biological organisms [40]. Through meticulous analysis of muscle arrangements, tendon structures, and bone geometries, researchers aim to replicate the biomechanical properties of natural limbs in prosthetic designs [41,42]. Recent advancements in imaging technologies, such as high-resolution MRI and CT scans, have facilitated more detailed and accurate anatomical reconstructions, providing prosthetists with invaluable insights into the complex interplay of musculoskeletal components [43]. Furthermore, understanding proprioceptive and tactile feedback mechanisms inherent in biological systems is crucial for enhancing user experience and functionality in prosthetic devices [44].

Proprioception, the sense of limb position and movement, plays a critical role in motor control and coordination. By integrating sensors capable of detecting joint angles, muscle contractions, and pressure distribution, prosthetic devices can provide users with real-time feedback, enhancing their ability to interact intuitively with the device and their environment [21]. Moreover, advancements in neural interface technologies have enabled researchers to develop prosthetic control mechanisms that closely mimic natural motor control strategies [8]. By deciphering the neural pathways and neural coding principles underlying voluntary movement, researchers can design intuitive and responsive interfaces that allow users to seamlessly manipulate their prosthetic limbs [21]. Neural interfaces, such as implanted electrodes or non-invasive brain-computer interfaces, enable direct communication between the prosthetic device and the user’s nervous system, facilitating naturalistic and intuitive control [45]. A profound understanding of biological systems serves as the foundation for biomimetic prosthetic design, enabling the development of devices that not only restore lost functionality but also provide users with a sense of natural movement and interaction [21]. Moreover, this understanding facilitates the exploration of advanced materials and manufacturing techniques, ensuring that biomimetic prosthetics achieve optimal performance and usability.

In biomimetic prosthetic design, a thorough understanding of biological systems is crucial. This involves analyzing musculoskeletal components, sensory feedback mechanisms, and control systems. Advanced imaging technologies aid in detailed anatomical reconstructions, while proprioceptive and tactile feedback mechanisms enhance user experience. Neural interface technologies enable intuitive control. This understanding forms the foundation for developing prosthetic devices that closely mimic natural limbs, ensuring optimal performance and usability.

### 2.2. Advanced Materials and Manufacturing Techniques

Advancements in materials science and additive manufacturing techniques have profoundly influenced the fabrication of prosthetic devices, imbuing them with biomimetic features that closely emulate the anatomical structure and mechanical properties of biological tissues [46]. This section explores the latest innovations in materials and manufacturing techniques, elucidating their impact on prosthetic design and functionality. A pivotal advancement in prosthetic fabrication involves the creation of anatomically-shaped sockets, crucial interfaces between the residual limb and the prosthetic device [47]. Leveraging 3D scanning technologies and computational modeling, prosthetists can craft custom-fit sockets that intricately match the contours of the user’s residual limb [48]. This personalized approach minimizes discomfort and bolsters stability, thereby enhancing overall user experience and prosthetic function [18]. Moreover, the adoption of lightweight and resilient materials, such as carbon fiber composites and elastomers, has significantly augmented the performance and durability of prosthetic components [49]. These materials exhibit exceptional strength-to-weight ratios and flexibility, enabling the creation of prosthetic devices capable of withstanding daily rigors while closely approximating the mechanical properties of natural tissues [50]. Additionally, additive manufacturing techniques, notably 3D printing, offer unparalleled versatility in prosthetic design and production [51]. By enabling the realization of intricate geometries and complex structures, additive manufacturing empowers researchers to push the boundaries of biomimetic prosthetic design [27]. From intricately articulated joints to bio-inspired textures, additive manufacturing facilitates the development of prosthetic devices that seamlessly integrate with the user’s body, fostering naturalistic movement [52].

The synergy between advancements in materials science and additive manufacturing techniques has enabled the fabrication of prosthetic devices endowed with biomimetic features, thereby advancing the pursuit of prosthetic limbs closely mirroring the structure and functionality of natural counterparts [21]. By deeply understanding biological systems, including musculoskeletal components, sensory feedback mechanisms, and control systems, researchers can create prosthetic devices that closely mimic natural limbs. This understanding enables the development of prosthetic limbs with improved aesthetics, functionality, and user experience. Moreover, integration with biological systems facilitates the implementation of advanced materials and manufacturing techniques, ensuring the creation of prosthetic components that exhibit biomimetic features. This approach not only restores lost functionality but also provides users with a sense of natural movement and interaction, ultimately enhancing the effectiveness and acceptance of prosthetic devices. Table 1 delineates the advancements in biomimetic prosthetics alongside their applications and associated limitations. Recent trends in biomimetics for developing bio-inspired prosthetics encompass a diverse array of innovations, with soft robotics drawing inspiration from soft-bodied organisms, resulting in prosthetics with enhanced flexibility and adaptability, exemplified by the octopus-inspired soft robotic gripper [53]. Novel mechanisms, such as the Variable Stiffness Parallel Elastic Actuation (VSPEA), are revolutionizing prosthetic knee joints, crucial for lower limb rehabilitation [28]. The Hannes hand exhibits human-like kinematic behavior and robust grasping, making it suitable for prosthetic applications requiring natural movement and a strong grip [54]. Investigations into prosthetic foot flexibility aim to optimize design and selection for improved user comfort and energy efficiency [55]. Advanced robotics hands offer distinct sensing capabilities, facilitating precise manipulation tasks [56]. Biomimetic approaches, including passive prosthetic spring-loaded knees [57], nanocrack-based electronic mechanosensors [58], and stretchable sensory-neuromorphic systems [59], hold promise for intelligent wearable electronics and health monitoring systems. Additionally, laboratory wear testers aid in evaluating the longevity and performance of dental materials essential for prosthetic restorations [60]. Enhanced flexible wearable electronic devices enable synchronization with human articulations, advancing mechanical prosthetics’ integration with natural movements [61].

Therefore, biomimetic design strategies in prosthetic limb development encompass a comprehensive approach rooted in understanding and replicating biological systems. Researchers aim to fabricate prosthetic devices that closely emulate natural limbs by exploring the intricate interaction of musculoskeletal components, sensory feedback mechanisms, and control systems in living organisms. Biomimetic approaches include bionic limbs with muscle-like actuators, sensory feedback systems, biomorphic design principles, bio-inspired control strategies, energy harvesting and storage mechanisms, adaptive materials and interfaces, tendon-driven actuation systems, and bio-inspired skin and texture. These strategies enable prosthetic limbs to achieve lifelike movement, enhance user perception and control, improve aesthetics and functionality, and provide naturalistic feedback, ultimately bridging the gap between artificial devices and natural appendages. Understanding biological systems is crucial for developing prosthetic devices that restore lost functionality and provide users with a sense of natural movement and interaction. Recent advancements in materials science and additive manufacturing techniques have further enhanced prosthetic design and functionality by enabling the creation of anatomically-shaped sockets, lightweight and resilient materials, and intricate prosthetic components. By deeply understanding biological systems and integrating with advanced materials and manufacturing techniques, researchers can create prosthetic devices that exhibit biomimetic features, enhancing the effectiveness and acceptance of prosthetic limbs. Recent trends in biomimetics for developing bio-inspired prosthetics encompass a diverse array of innovations, including soft robotics, variable stiffness parallel elastic actuation, human-like kinematic behavior in robotic hands, optimized prosthetic foot flexibility, and intelligent wearable electronics for health monitoring systems, among others.

Advancements in materials science and additive manufacturing techniques have revolutionized prosthetic device fabrication, fostering biomimetic features mirroring natural tissues. Leveraging 3D scanning and computational modeling, custom-fit sockets optimize user comfort and stability. Lightweight carbon fiber composites and elastomers enhance durability and mimic natural tissue properties. Additive manufacturing, notably 3D printing, enables intricate prosthetic designs, facilitating naturalistic movement. These synergistic advancements advance prosthetic limb development, bridging the gap between artificial and natural appendages. Through deep understanding of biological systems and integration with advanced materials and manufacturing techniques, researchers create prosthetic devices with biomimetic features, enhancing functionality and user acceptance. Recent trends encompass soft robotics, variable stiffness actuation, human-like robotic hands, optimized foot flexibility, and wearable electronics for health monitoring, reflecting a diverse landscape of biomimetic innovations in prosthetic design.

## 3. Integration with Biological Systems

The integration of prosthetic devices with biological systems represents a pivotal frontier in prosthetic technology, aiming to enhance compatibility, functionality, and user experience. This section explores two key avenues of integration: osseointegration and neural interfaces with sensory feedback. Through innovative approaches such as direct bone integration and bidirectional communication with the nervous system, researchers are advancing the field of prosthetics towards more naturalistic and intuitive solutions [62]. By delving into the intricate interplay between artificial devices and biological organisms, these advancements offer promising prospects for improving the quality of life for individuals with limb loss [5].

### 3.1. Osseointegration

Integration of prosthetic devices with biological systems represents a significant trend aimed at enhancing compatibility and functionality [63]. Osseointegration, a pioneering approach in this regard, involves the direct integration of the prosthetic limb with the user’s bone, offering several advantages over traditional socket-based prosthetics [12]. Recent advancements in osseointegration techniques have focused on improving the biomechanical interface between the prosthetic implant and the residual bone [11]. Innovations in implant design, surface modifications, and surgical techniques have contributed to enhanced bone-implant integration, resulting in improved stability and comfort for users [64]. Furthermore, the use of advanced biomaterials, such as titanium alloys and biocompatible coatings, has reduced the risk of implant rejection and infection, ensuring long-term success and durability [65].

One notable recent development in osseointegration is the emergence of percutaneous osseointegrated prostheses (POP), which feature a transcutaneous implant that penetrates the skin and directly interfaces with the underlying bone [66,67]. This approach eliminates the need for a traditional socket interface, reducing issues related to skin irritation, discomfort, and restricted range of motion. Additionally, POP systems offer greater mechanical stability and proprioceptive feedback, enabling users to achieve more natural movement patterns and improved functional outcomes [68]. Therefore, osseointegration represents a cutting-edge approach to prosthetic integration with biological systems, offering enhanced stability, comfort, and functionality compared to traditional socket-based prosthetics. Recent advancements in implant design, biomaterials, and surgical techniques continue to drive improvements in osseointegration outcomes, paving the way for more widespread adoption and improved quality of life for individuals with limb loss. Overall, osseointegration offers enhanced stability, comfort, and functionality compared to traditional methods, driving improvements in prosthetic integration and quality of life for individuals with limb loss.

### 3.2. Neural Interfaces and Sensory Feedback

Advancements in neural interfaces and sensory feedback systems have ushered in a new era of prosthetic technology, enabling more intuitive control and natural movement of prosthetic limbs [69,70]. These technologies bridge the gap between man-made devices and biological systems, allowing users to experience tactile feedback and proprioceptive information, thereby enhancing their ability to interact with their environment. Recent developments in neural interfaces have focused on improving the bidirectional communication between prosthetic devices and the user’s nervous system [71]. Intracortical electrodes, for example, enable direct neural recordings from the motor cortex, allowing users to control prosthetic limbs with unprecedented precision and accuracy [72]. Moreover, advancements in signal processing algorithms and machine learning techniques have enhanced the decoding of neural signals, enabling more intuitive and natural movement of prosthetic limbs [73]. In addition to motor control, recent efforts have also been directed towards providing sensory feedback to prosthetic users. Tactile sensors embedded in prosthetic hands, for instance, enable users to perceive the texture, shape, and hardness of objects they interact with [74]. Similarly, proprioceptive feedback systems provide users with information about the position and movement of their prosthetic limbs, enhancing their sense of embodiment and motor control.

A significant recent development in sensory feedback is the integration of artificial skin with prosthetic devices, capable of sensing and transmitting tactile stimuli to the user’s residual limb [62]. These artificial skin systems utilize flexible electronics and tactile sensors to mimic the sensitivity and responsiveness of human skin, providing users with a rich sensory experience during prosthetic use. In summary, advancements in neural interfaces and sensory feedback systems hold tremendous potential for enhancing the functionality and usability of prosthetic limbs. Recent developments in intracortical electrodes, tactile sensors, and artificial skin technologies offer new opportunities for improving user experience and restoring sensory-motor capabilities in individuals with limb loss [75]. By integrating with biological systems and providing naturalistic feedback, these technologies pave the way for more seamless integration of prosthetic devices into daily life [76].

Therefore, the integration of prosthetic devices with biological systems represents a significant advancement in prosthetic technology, aiming to enhance compatibility, functionality, and user experience. This section delves into two pivotal avenues of integration: osseointegration and neural interfaces with sensory feedback. Osseointegration involves directly integrating the prosthetic limb with the user’s bone, offering advantages over traditional socket-based prosthetics. Recent advancements focus on improving the biomechanical interface, resulting in enhanced stability and comfort for users, exemplified by the emergence of POP. On the other hand, advancements in neural interfaces and sensory feedback systems enable more intuitive control and natural movement of prosthetic limbs, bridging the gap between artificial devices and biological systems. These technologies facilitate precise control through bidirectional communication with the nervous system and provide rich sensory experiences, offering promising prospects for improving user experience and restoring sensory-motor capabilities in individuals with limb loss. Table 2 summarizes the key aspects of osseointegration and neural interfaces with sensory feedback in the context of prosthetic technology. Each aspect is presented with relevant information from the provided text, facilitating easy comparison between the two integration methods.

Advancements in neural interfaces and sensory feedback systems revolutionize prosthetic technology, enabling more intuitive control and natural movement. These technologies bridge artificial devices with biological systems, providing tactile and proprioceptive feedback for enhanced interaction with the environment. Recent developments focus on bidirectional communication with the nervous system, improving prosthetic limb control precision. Tactile sensors in prosthetic hands and artificial skin integration offer rich sensory experiences, improving functionality and usability. These advancements hold promise for restoring sensory-motor capabilities in individuals with limb loss and integrating prosthetic devices seamlessly into daily life.

## 4. Customization and Personalization

In the domain of prosthetic design, customization and personalization are crucial pathways that drive innovation and prioritize user-centric solutions [77]. It is paramount to acknowledge the transformative influence of tailored prosthetic devices, particularly in light of advancements in biomimetics and additive manufacturing techniques that have reshaped the field. By tailoring prosthetic solutions to harmonize with the unique anatomical features and lifestyle preferences of each individual, prosthetists not only enhance comfort and functionality but also cultivate a renewed sense of ownership and confidence among users [18]. Through the seamless integration of state-of-the-art technologies and collaborative design processes, the era of personalized prosthetics marks a significant advancement in prosthetic care, offering unparalleled levels of satisfaction and enhancing the overall quality of life for users.

### 4.1. Tailored Solutions

Advancements in biomimetics have sparked a significant transformation in the realm of prosthetic devices, leading to a paradigm shift towards highly customized and personalized solutions tailored to meet the unique needs and preferences of individual users [78]. Unlike traditional one-size-fits-all approaches, this new trend leverages cutting-edge technologies to develop prosthetic devices that seamlessly integrate with the user’s anatomy and lifestyle [79]. A cornerstone of this customization is the adoption of 3D scanning technology, which allows prosthetists to obtain precise and comprehensive digital representations of the user’s residual limb [80]. By capturing the unique contours and dimensions of the residual limb, 3D scanning serves as the foundation for developing custom-fitted prosthetic sockets that optimize both comfort and functionality [81]. Furthermore, computer-aided design (CAD) software empowers prosthetists to manipulate and refine these digital models, enabling fine-tuning of the prosthetic design to align precisely with the user’s specific requirements and preferences [82]. Recent advancements in 3D printing technology have further revolutionized the customization process by facilitating rapid prototyping and iterative design enhancements [83]. Prosthetic components can now be manufactured with intricate geometries and complex structures, tailored to the individual user’s needs with unparalleled precision and accuracy [84]. Moreover, the utilization of advanced materials, such as biocompatible polymers and lightweight alloys, enables the creation of prosthetic devices that not only offer durability but also blend seamlessly with the user’s aesthetic preferences. In addition to anatomical customization, prosthetists also take into account the user’s lifestyle and functional requirements when designing prosthetic solutions [18]. It acknowledges that different users have diverse needs based on their lifestyle, activities, and personal preferences. For instance, individuals with active lifestyles, such as athletes or outdoor enthusiasts, may demand prosthetic devices optimized for sports and recreational activities. These devices need to be durable, lightweight, and provide enhanced performance to withstand rigorous physical activities.

On the other hand, some users may prioritize comfort and appearance for everyday use. They may seek prosthetic solutions that offer a natural look, fit comfortably, and allow for ease of movement during daily tasks. Understanding these varying needs is crucial for prosthetists to provide tailored solutions that meet each user’s specific requirements. To achieve this level of customization, prosthetists engage in close collaboration with users throughout the design process. By actively soliciting feedback and incorporating user input, prosthetists ensure that the final device addresses the user’s concerns and preferences effectively. This collaborative approach enables prosthetists to create prosthetic devices that are uniquely tailored to each individual, enhancing user satisfaction and overall quality of life [82]. Advancements in biomimetics have played a significant role in facilitating this customization and personalization of prosthetic devices. Biomimetic design principles draw inspiration from nature to create prosthetic solutions that closely mimic the functionality and aesthetics of natural limbs. By integrating advanced technologies such as 3D scanning, computer-aided design, and 3D printing, prosthetists can translate these biomimetic concepts into highly customized prosthetic devices. These technologies allow for precise and detailed mapping of the user’s anatomy, enabling prosthetists to create prosthetic components that closely align with the user’s specific physiological characteristics [83]. This level of precision ensures optimal fit, comfort, and functionality, ultimately enhancing the user’s overall satisfaction and quality of life.

Customization and personalization are integral aspects of prosthetic design, driven by advancements in biomimetics and additive manufacturing. Tailoring prosthetic solutions to individual anatomical features and lifestyle preferences enhances comfort, functionality, and user confidence. Leveraging 3D scanning technology, prosthetists obtain precise digital representations of the residual limb, enabling custom-fitted prosthetic sockets. CAD software refines digital models, allowing precise alignment with user requirements. Three-dimensional printing facilitates rapid prototyping, enabling intricate designs tailored to individual needs. Advanced materials ensure durability and aesthetic integration. Lifestyle considerations, such as activity level, guide prosthetic design, ensuring suitability for various users. Close collaboration between prosthetists and users throughout the design process ensures that devices meet specific requirements effectively. Biomimetic principles inspire customized solutions that mimic natural limbs’ functionality and aesthetics, enhancing user satisfaction and quality of life.

### 4.2. Enhanced Comfort and Functionality

The customization and personalization of prosthetic devices represent significant advancements that have led to improvements in both aesthetics and functionality for users. This approach tailors prosthetic solutions to match the unique anatomy and lifestyle of each individual, allowing prosthetists to address specific challenges and optimize performance in real-world scenarios [18]. One of the primary benefits of customization is the optimization of socket fit, which is crucial for ensuring comfort and stability during prosthetic use. Traditional socket-based prosthetics often encounter issues such as pressure sores, discomfort, and poor suspension due to ill-fitting sockets that fail to distribute load adequately and accommodate variations in residual limb shape [85]. However, through techniques such as 3D scanning and computer-aided design, prosthetists can now create custom-fitted sockets that closely conform to the user’s residual limb, minimizing pressure points and enhancing overall comfort [86].

Furthermore, customization enables prosthetists to incorporate features and functionalities tailored to the user’s specific needs and preferences. For instance, individuals with above-knee limb loss may benefit from prosthetic knees equipped with microprocessor-controlled hydraulic systems, offering enhanced stability and control during various activities [87]. Similarly, users with upper limb loss may opt for prosthetic hands with modular attachments, providing versatile functionality tailored to different tasks and activities. Apart from functional enhancements, customization also plays a crucial role in improving the aesthetic appeal of prosthetic devices [88]. By integrating color, texture, and design elements reflecting the user’s personality and style, prosthetists promote a sense of ownership and self-expression, fostering a positive body image and boosting self-confidence [89]. Overall, the customization and personalization of prosthetic devices contribute to enhanced comfort, functionality, and user satisfaction [90]. Leveraging advancements in biomimetics and additive manufacturing, prosthetists can now create prosthetic solutions that not only restore lost function but also empower users to lead active and fulfilling lives. This integration of cutting-edge technologies allows for highly customized prosthetic devices that closely mimic the form and function of natural limbs, ultimately enhancing the overall quality of life for prosthetic users.

Therefore, customization and personalization are essential in prosthetic design, driving innovation and user-centric solutions. Advancements in biomimetics have led to highly tailored prosthetic solutions, departing from traditional one-size-fits-all approaches and leveraging technologies like 3D scanning and printing to seamlessly integrate with users’ anatomy and lifestyle. Tailored prosthetic sockets optimize comfort and functionality, while consideration of users’ specific needs and preferences ensures devices meet individual requirements. This collaborative approach between prosthetists and users enhances overall satisfaction and quality of life. Additionally, customization improves both aesthetics and functionality, addressing challenges such as pressure sores and enhancing stability during prosthetic use. Advanced features like microprocessor-controlled hydraulic systems and modular attachments further enhance functionality, while aesthetic customization promotes positive body image and self-confidence. Therefore, customization and personalization contribute to enhanced comfort, functionality, and user satisfaction, empowering prosthetic users to lead fulfilling lives with prosthetic devices closely aligned with their needs and preferences. Further, researchers are investigating biomimetic prosthetic limbs with self-repairing materials, inspired by biological systems [91,92]. These materials mimic natural healing mechanisms, aiming to improve limb durability and longevity [93]. By detecting and repairing damage autonomously, they reduce maintenance needs and enhance user satisfaction. This research promises more resilient and efficient prosthetic solutions for individuals with limb loss.

### 4.3. Self-Repairing Materials in Biomimetic Prosthetic Limbs

Self-repairing materials represent an intriguing area of research at the intersection of biomimetics and the development of bio-inspired prosthetic limbs [93]. These materials offer the remarkable ability to autonomously repair damage, akin to the self-healing mechanisms observed in biological organisms [94,95]. One prominent approach involves the integration of microcapsules or vascular networks filled with healing agents into the material matrix [96]. In the event of damage, these capsules rupture or the vascular networks break, releasing the healing agents to fill and mend the affected area. This concept draws inspiration from biological processes such as blood clotting and wound healing, where the body mobilizes healing agents to repair tissue damage.

Another strategy for self-repairing materials involves utilizing reversible chemical reactions or physical interactions within the material structure [97]. When damage occurs, these reactions or interactions are triggered, facilitating the restoration of the material’s integrity. Dynamic bonds or polymers capable of undergoing reversible changes in response to external stimuli are commonly employed in this approach [98]. Furthermore, researchers are exploring the integration of sensors and actuators within self-repairing materials, enabling them to detect damage and initiate repair processes autonomously [99]. This real-time monitoring and repair capability closely mimic the sensory feedback and self-repair mechanisms observed in natural organisms [99,100]. By integrating sensors, these materials can detect changes in their environment and respond accordingly, enhancing their ability to detect and repair damage in prosthetic limbs.

Assessing the cytotoxicity of innovative prosthetic materials is crucial for ensuring their safety and biocompatibility when in contact with the body’s cells. This aspect is essential for the successful integration and long-term use of prosthetic limbs in individuals. Therefore, investigating the cytotoxicity of prosthetic components is an important step in their development. For example, research by Pranczk et al. [101] provides valuable insights into cytotoxicity assessment methods. Understanding cytotoxicity profiles can aid in the selection of materials that minimize adverse reactions and promote tissue compatibility, ultimately enhancing the safety and effectiveness of prosthetic devices. Incorporating cytotoxicity studies into the development process of biomimetic prosthetic limbs ensures that the materials used are not only capable of self-repair but also biocompatible and safe for prolonged contact with biological tissues. This holistic approach to prosthetic design promotes the development of innovative solutions that prioritize both functionality and biological compatibility, thereby improving the overall quality of life for prosthetic users.

## 5. Conclusions and Future Prospective

In conclusion, biomimetics is guiding revolutionary advancements in prosthetic limb development, ushering in an era where bio-inspired prosthetics closely emulate the intricate form and function of natural limbs. By strategically employing biomimetic design strategies and integrating with biological systems, researchers and prosthetists are reshaping prosthetic technology, profoundly impacting the lives of individuals with limb loss or impairment. The profound insights gained from studying biological systems, including musculoskeletal structures, sensory feedback mechanisms, and control systems, have paved the way for groundbreaking prosthetic designs prioritizing naturalistic movement and seamless interaction. Integration with biological systems, such as osseointegration and neural interfaces, marks a new epoch of prosthetic functionality, offering users enhanced stability, control, and sensory feedback.

Moreover, the pursuit of customization and personalization has provided prosthetic users with tailored solutions optimizing comfort, functionality, and aesthetic appeal. Leveraging technologies like 3D scanning, computer-aided design, and additive manufacturing, prosthetists create devices that harmonize seamlessly with the user’s anatomy and lifestyle, fostering heightened satisfaction and quality of life. Looking ahead, the horizon of biomimetic prosthetics holds boundless promise, propelled by ongoing research and interdisciplinary collaboration. Continued advancements in materials science, robotics, neural interfaces, and sensor technologies will unveil prosthetic devices with unprecedented functionality, realism, and integration. Interdisciplinary partnerships between researchers, clinicians, engineers, and prosthetic users will nurture a holistic approach to prosthetic development, ensuring that future innovations cater adeptly to diverse user needs and preferences. In summary, biomimetics is driving profound metamorphosis in prosthetic limb development, aiming to replicate the elegance and efficiency of natural biological systems. As the field evolves, bio-inspired prosthetics will play a pivotal role in augmenting mobility, fostering independence, and enriching the quality of life for individuals with limb loss or impairment. This vision foresees a future where prosthetic technology transcends restoration to enable seamless integration with the human body.

## Figures and Tables

**Figure 1 biomimetics-09-00273-f001:**
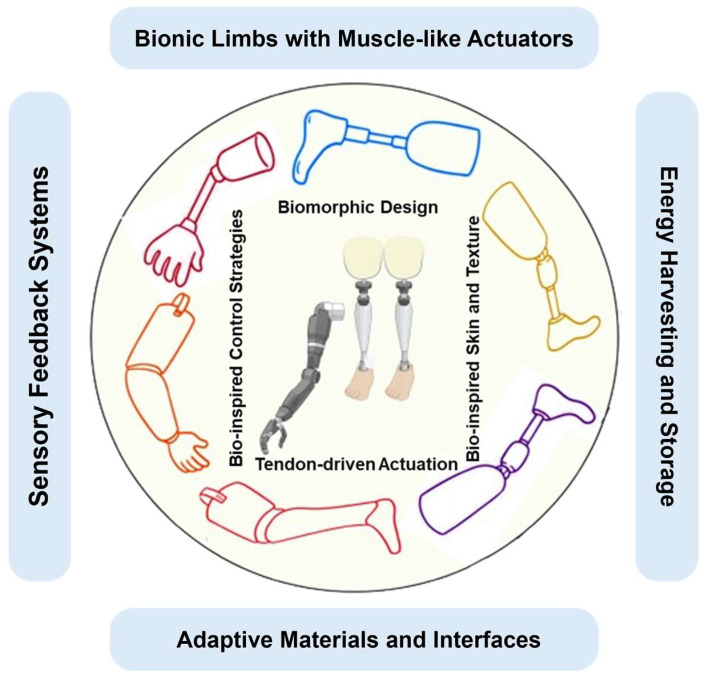
Some biomimetic approaches in the development of bio-inspired prosthetic limbs: (Bionic limbs with muscle-like actuators: Prosthetic limbs incorporate actuators resembling natural muscles for lifelike movement. Sensory feedback systems: Integration of systems providing touch and proprioceptive feedback for enhanced perception and control. Biomorphic design: Mimicking natural limb structure and function to create prosthetic limbs with improved aesthetics and functionality. Bio-inspired control strategies: Implementing control algorithms based on biological neural systems for intuitive movement. Energy harvesting and storage: Utilizing mechanisms inspired by nature to harvest and store energy, enhancing prosthetic autonomy. Adaptive materials and interfaces: Incorporating materials and interfaces that adapt to user needs and environmental changes. Tendon-driven actuation: Employing tendon-driven systems to transmit motion, mimicking natural limb movement. Bio-inspired skin and texture: Creating prosthetic surfaces resembling biological skin for a lifelike appearance and tactile experience).

**Table 1 biomimetics-09-00273-t001:** Trends in biomimetics for developing bio-inspired prosthetics.

Type of Prosthesis	Description/Mechanism	Application	Reference
Soft robotics	Drawing inspiration from soft-bodied organisms to create prosthetics with enhanced flexibility, adaptability, and resilience.	Octopus-inspired soft robotic gripper.	[53]
Prosthetic knee joint	Introducing a novel Variable Stiffness Parallel Elastic Actuation (VSPEA) mechanism for an active prosthetic knee joint.	Prosthetic knee joints are vital for lower limb rehabilitation, enhancing the lives of individuals with disabilities by providing essential functions like stance support and swing actuation.	[28]
Hannes hand	Kinematic analysis was employed to investigate the extent to which the Hannes hand exhibits human-like synergistic kinematic behavior. Additionally, grip robustness, akin to human grasping, was assessed as another factor.	The Hannes hand mimics human-like kinematic behavior and robust grasping, making it suitable for prosthetic applications requiring natural movement and strong grip.	[54]
Prosthetic foot	Investigating the impact of prosthetic foot forefoot flexibility on both the oxygen cost and subjective preference rankings of individuals with unilateral transtibial prostheses.	The application involves leveraging prosthetic foot forefoot flexibility to optimize design and selection, thereby enhancing user comfort, mobility, and energy efficiency for individuals with unilateral and bilateral transtibial prostheses.	[55]
Robotics hands	Exploring the current advancements in dexterous robotics end-effectors, commonly referred to in the literature as “robotic hands” or “dexterous multi-fingered” robot hands.	Multi-fingered hands offer a distinct sensing capability. The collaboration of position, force, tactile, and proximity sensors presents an opportunity to gather information regarding the mechanical and physical characteristics of objects and tasks.	[56]
Passive prosthetic spring-loaded knee	Modeling a pneumatic-controlled biomimetic articulated passive prosthetic spring-loaded knee mechanism for transfemoral amputees.	Facilitating prosthetic devices to improve quality of life and standards of living.	[57]
Nanocrack-based electronic whisker-type mechanosensor	Achieves unparalleled sensitivity by leveraging nanocracks within its structure to detect mechanical stimuli, enabling precise perception of subtle forces and surface morphology with a resolution down to 30 nm.	Wearable health monitoring systems and human-machine interfaces. Its integration into wearable smart systems enables remote monitoring of elderly individuals’ posture and movements, enhancing safety.	[58]
Bioinspired stretchable sensory-neuromorphic system	The system integrates a stretchable capacitive pressure sensor (artificial mechanoreceptor), resistive random-access memory (artificial synapse), and quantum dot light-emitting diode (epidermal photonic actuator) into a rigid-island structure with a sinter-free printable conductor.	Offers groundbreaking potential in intelligent wearable electronics, particularly in prosthetics with advanced sensory and actuating capabilities.	[59]
Laboratory wear tester utilizing a ball-on-3-specimen configuration with a rotating zirconia sphere as the hard antagonist.	The wear rates of dental materials are determined by quantifying scar dimensions, demonstrating zirconia ceramics as having the lowest wear rates, followed by feldspathic ceramic and ceramic-polymer composite, with lithium disilicate displaying the highest wear rate.	Assisting in evaluating their potential longevity under conditions mimicking basic occlusal contact, essential for preventing severe material loss and premature failure of natural teeth or prosthetic restorations.	[60]
Advanced flexible wearable electronic devices	This device enhances current flexible wearable electronic devices by integrating microscale materials and a biomimetic stretch optimization approach, resulting in a flexible sensory device with improved mechanical strength and elongation capacity, as well as notable enhancements in stretchability.	This innovation enables the synchronization and emulation of extensive tensile movements between mechanical prosthetics and human articulations.	[61]

**Table 2 biomimetics-09-00273-t002:** Key aspects of osseointegration and neural interfaces with sensory feedback in the context of prosthetic technology.

Aspect/Parameter	Osseointegration	Neural Interfaces and Sensory Feedback
Definition	Direct integration of prosthetic limb with user’s bone.	Bridging the gap between man-made devices and biological systems.
Advantages	Enhanced stability, comfort, and functionality.	More intuitive control and natural movement of prosthetic limbs.
Recent developments	Emergence of percutaneous osseointegrated prostheses (POP).	Integration of artificial skin with prosthetic devices for tactile feedback.
Key features	Transcutaneous implant interfaces directly with underlying bone.	Intracortical electrodes enable direct neural recordings from motor cortex.
Materials utilized	Titanium alloys, biocompatible coatings.	Flexible electronics, tactile sensors for mimicking human skin sensitivity.
Improvement focus	Biomechanical interface between prosthetic implant and residual bone.	Bidirectional communication between prosthetic devices and user’s nervous system.
Applications	Enhanced stability, comfort, and functionality in prosthetic limbs.	Perception of texture, shape, hardness of objects with tactile sensors in prosthetic hands.
Disadvantages	Risk of infection, potential for implant rejection.	Complexity of surgical procedures, limited availability of advanced sensory feedback systems.
Future prospects	Widespread adoption and improved quality of life for individuals with limb loss.	Restoring sensory-motor capabilities, seamless integration of prosthetic devices into daily life.

## Data Availability

No new data were created or analyzed in this study. Data sharing is not applicable to this article.

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
