# Peer review of "Recent Advances in Biomimetics for the Development of Bio-Inspired Prosthetic Limbs"

_biomimetics, 2024, doi:10.3390/biomimetics9050273_

Round 1

Reviewer 1 Report

Comments and Suggestions for Authors

The paper is interesting and relevant. It discusses a new field of knowledge called Biomimetics. Biomimetics encompasses an interdisciplinary approach that draws on biology, materials science, robotics, and bioengineering knowledge. Biomimetics, which is called "nature-inspired engineering," involves meticulously studying and emulating biological systems to tackle complex human challenges. Biomimetic prosthetics have shown promising results in a variety of applications. In this paper, the authors have provided a comprehensive overview of the latest trends and innovations driving the development of biomimetic prosthetics. The matter is presented in a comprehensive manner. That is why the information provided in this review is interesting and relevant.

The ongoing hostilities in Ukraine, Israel, and the Gaza Strip have led to an increased demand for prosthetic limbs. This is due to the fact that the main types of injuries are those caused by mine blasts, missiles, artillery, and bomb attacks. Therefore, the information presented in this article is extremely relevant for developing prostheses that restore lost functionality and provide users with a sense of natural movement and interaction.

The structure of the paper is rational. The paper contains the main sections, such as "1. Introduction", "2. Biomimetic design strategies", "3. Integration with biological systems", "4. Customization and personalization", and "5. Conclusions and future prospective". The list of references contains a list of modern publications within the last five years. All cited publications are relevant to the field of study and complement each other. All the references provided are applicable and sufficient. The text of the paper is systematized and well-structured. The Abstract and Conclusions fully reflect all the main results of this article. In my opinion, the new result of this study is Table 1. In Table 1, the authors have presented the current advances in biomimetic prosthetics, as well as their applications and associated limitations. The information in Table 1 systematizes the current body of knowledge in this field.

This paper is up-to-date and relevant, contributing to the body of knowledge. I have no comments on its content or the review results presented in it. I suggest that the paper be published in its current form in the Biomimetics Journal.

Author Response

Reviewer #1.

The paper is interesting and relevant. It discusses a new field of knowledge called Biomimetics. Biomimetics encompasses an interdisciplinary approach that draws on biology, materials science, robotics, and bioengineering knowledge. Biomimetics, which is called "nature-inspired engineering," involves meticulously studying and emulating biological systems to tackle complex human challenges. Biomimetic prosthetics have shown promising results in a variety of applications. In this paper, the authors have provided a comprehensive overview of the latest trends and innovations driving the development of biomimetic prosthetics. The matter is presented in a comprehensive manner. That is why the information provided in this review is interesting and relevant.

The ongoing hostilities in Ukraine, Israel, and the Gaza Strip have led to an increased demand for prosthetic limbs. This is due to the fact that the main types of injuries are those caused by mine blasts, missiles, artillery, and bomb attacks. Therefore, the information presented in this article is extremely relevant for developing prostheses that restore lost functionality and provide users with a sense of natural movement and interaction.

The structure of the paper is rational. The paper contains the main sections, such as "1. Introduction", "2. Biomimetic design strategies", "3. Integration with biological systems", "4. Customization and personalization", and "5. Conclusions and future prospective". The list of references contains a list of modern publications within the last five years. All cited publications are relevant to the field of study and complement each other. All the references provided are applicable and sufficient. The text of the paper is systematized and well-structured. The Abstract and Conclusions fully reflect all the main results of this article. In my opinion, the new result of this study is Table 1. In Table 1, the authors have presented the current advances in biomimetic prosthetics, as well as their applications and associated limitations. The information in Table 1 systematizes the current body of knowledge in this field.

This paper is up-to-date and relevant, contributing to the body of knowledge. I have no comments on its content or the review results presented in it. I suggest that the paper be published in its current form in the Biomimetics Journal.

Res: We thank Reviewer #1 for their positive evaluation and endorsement of our manuscript. We appreciate their recognition of its relevance, structure, and contributions to the field. We will ensure that any final revisions align with the standards of the journal.

Reviewer 2 Report

Comments and Suggestions for Authors

Authors described the latest trends and innovations driving the development of biomimetic prosthetics. By examining recent breakthroughs in materials science, robotics, neural interfaces, and sensor technologies, the review aims to elucidate the  transformative potential of biomimetic design principles in prosthetic limb development. Moreover, the article shows case studies and real-world applications where biomimetic prosthetics have demonstrably improved the lives of individuals with limb loss, thereby highlighting the importance of this field in rehabilitation and assistive technologies. The review article is well-designed. The entire work is based on a large number of literature references that are relatively recent. The added value of any review paper should be a new perspective, analyzing or synthesizing information in a way that leads to new insights or conclusions. Unfortunately, it is difficult to find such elements in the present work. I feel that the work will be suitable for publication as a paper in Biomimetics journal after few major  corrections. Some comments shown below could help authors to improve the work.

1) Please share your opinion on the subject matter discussed. It is recommended to include a paragraph expressing your opinion for each subsection.

2) The topic of the paper is closely related to self-repairing materials, and I believe it would be beneficial to include information on this topic.

3) The conclusions and future perspectives should encompass the primary advantages and disadvantages of the biomimetic solutions employed.

4) I think it would be good to include in this prayc a diagram showing the path from laboratory work to industrial/commercial scale transition regarding biomimetic prosthesis applications. Here it would also be useful to refer to the most recent patent literature.

5) In Table 1, the first column from the left should be labeled with a name, such as 'Type of Prosthesis', as it does not currently indicate a trend.

6) An important aspect in developing the use of an innovative prosthesis is to study, among other things, the cytotoxicity of its components when in contact with the body's cells. Therefore, it would be beneficial to address this issue, which can be supported by the publication mentioned here: e.g. "Pranczk, J., Jacewicz, D., Wyrzykowski, D., & Chmurzynski, L. (2014). Platinum (II) and palladium (II) complex compounds as anti-cancer drugs. Methods of cytotoxicity determination. Current Pharmaceutical Analysis, 10(1), 2-9."

Comments on the Quality of English Language

Minor editing of English language required.

Author Response

Reviewer #2

Authors described the latest trends and innovations driving the development of biomimetic prosthetics. By examining recent breakthroughs in materials science, robotics, neural interfaces, and sensor technologies, the review aims to elucidate the transformative potential of biomimetic design principles in prosthetic limb development. Moreover, the article shows case studies and real-world applications where biomimetic prosthetics have demonstrably improved the lives of individuals with limb loss, thereby highlighting the importance of this field in rehabilitation and assistive technologies. The review article is well-designed. The entire work is based on a large number of literature references that are relatively recent. The added value of any review paper should be a new perspective, analyzing or synthesizing information in a way that leads to new insights or conclusions. Unfortunately, it is difficult to find such elements in the present work. I feel that the work will be suitable for publication as a paper in Biomimetics journal after few major corrections. Some comments shown below could help authors to improve the work.

1) Please share your opinion on the subject matter discussed. It is recommended to include a paragraph expressing your opinion for each subsection.

Res: We appreciate Reviewer #2's thorough evaluation of our manuscript and their acknowledgment of its focus on the latest trends and innovations in biomimetic prosthetics. As suggested, we have revised the manuscript to include paragraphs expressing our opinions on the subject matter discussed in each subsection. We believe these additions enhance the overall value of the manuscript by providing additional insights and perspectives. Thank you for your valuable feedback and suggestions.

(Line 157)

In prosthetic limb development, biomimetic design is pivotal, aiming to replicate biological intricacies. By studying musculoskeletal dynamics, sensory feedback, and control mechanisms in organisms, researchers create prosthetics that mimic natural limbs. These include muscle-like actuators for lifelike movement and sensory feedback systems for enhanced control. Biomimetic principles drive aesthetic and functional improvements, integrating bio-inspired materials, energy harvesting, and tendon-driven actuation. Deep biological understanding and advanced manufacturing advance this field, bridging the gap between artificial and natural limbs, enhancing the lives of amputees

(Line 198)

In biomimetic prosthetic design, a thorough understanding of biological systems is crucial. This involves analyzing musculoskeletal components, sensory feedback mechanisms, and control systems. Advanced imaging technologies aid in detailed anatomical reconstructions, while proprioceptive and tactile feedback mechanisms enhance user experience. Neural interface technologies enable intuitive control. This understanding forms the foundation for developing prosthetic devices that closely mimic natural limbs, ensuring optimal performance and usability.

(Line 284)

Advancements in materials science and additive manufacturing techniques have revolutionized prosthetic device fabrication, fostering biomimetic features mirroring natural tissues. Leveraging 3D scanning and computational modeling, custom-fit sockets optimize user comfort and stability. Lightweight carbon fiber composites and elastomers enhance durability and mimic natural tissue properties. Additive manufacturing, notably 3D printing, enables intricate prosthetic designs, facilitating naturalistic movement. These synergistic advancements advance prosthetic limb development, bridging the gap between artificial and natural appendages. Through deep understanding of biological systems and integration with advanced materials and manufacturing techniques, researchers create prosthetic devices with biomimetic features, enhancing functionality and user acceptance. Recent trends encompass soft robotics, variable stiffness actuation, human-like robotic hands, optimized foot flexibility, and wearable electronics for health monitoring, reflecting a diverse landscape of biomimetic innovations in prosthetic design.

(Line 330)

Overall, osseointegration offers enhanced stability, comfort, and functionality compared to traditional methods, driving improvements in prosthetic integration and quality of life for individuals with limb loss.

(Line 382)

Advancements in neural interfaces and sensory feedback systems revolutionize prosthetic technology, enabling more intuitive control and natural movement. These technologies bridge artificial devices with biological systems, providing tactile and proprioceptive feedback for enhanced interaction with the environment. Recent developments focus on bidirectional communication with the nervous system, improving prosthetic limb control precision. Tactile sensors in prosthetic hands and artificial skin integration offer rich sensory experiences, improving functionality and usability. These advancements hold promise for restoring sensory-motor capabilities in individuals with limb loss and integrating prosthetic devices seamlessly into daily life.

(Line 449)

Customization and personalization are integral aspects of prosthetic design, driven by advancements in biomimetics and additive manufacturing. Tailoring prosthetic solutions to individual anatomical features and lifestyle preferences enhances comfort, functionality, and user confidence. Leveraging 3D scanning technology, prosthetists obtain precise digital representations of the residual limb, enabling custom-fitted prosthetic sockets. CAD software refines digital models, allowing precise alignment with user requirements. 3D printing facilitates rapid prototyping, enabling intricate designs tailored to individual needs. Advanced materials ensure durability and aesthetic integration. Lifestyle considerations, such as activity level, guide prosthetic design, ensuring suitability for various users. Close collaboration between prosthetists and users throughout the design process ensures that devices meet specific requirements effectively. Biomimetic principles inspire customized solutions that mimic natural limbs' functionality and aesthetics, enhancing user satisfaction and quality of life.

(Line 506)

Further, researchers are investigating biomimetic prosthetic limbs with self-repairing materials, inspired by biological systems [92.93]. These materials mimic natural healing mechanisms, aiming to improve limb durability and longevity [94]. By detecting and repairing damage autonomously, they reduce maintenance needs and enhance user satisfaction. This research promises more resilient and efficient prosthetic solutions for individuals with limb loss.

2) The topic of the paper is closely related to self-repairing materials, and I believe it would be beneficial to include information on this topic.

Res: As suggested, we have incorporated relevant information in the revised manuscript on self-repairing materials to provide a more comprehensive overview of biomimetic design principles in prosthetic limb development.

(Line 512)

4.3. Self-repairing materials in biomimetic prosthetic limbs

Self-repairing materials represent an intriguing area of research at the intersection of biomimetics and the development of bio-inspired prosthetic limbs [94]. These materials offer the remarkable ability to autonomously repair damage, akin to the self-healing mechanisms observed in biological organisms [95,96]. One prominent approach involves the integration of microcapsules or vascular networks filled with healing agents into the material matrix [97]. In the event of damage, these capsules rupture or the vascular networks break, releasing the healing agents to fill and mend the affected area. This concept draws inspiration from biological processes such as blood clotting and wound healing, where the body mobilizes healing agents to repair tissue damage.

Another strategy for self-repairing materials involves utilizing reversible chemical reactions or physical interactions within the material structure [98]. When damage occurs, these reactions or interactions are triggered, facilitating the restoration of the material's integrity. Dynamic bonds or polymers capable of undergoing reversible changes in response to external stimuli are commonly employed in this approach [99]. Furthermore, researchers are exploring the integration of sensors and actuators within self-repairing materials, enabling them to detect damage and initiate repair processes autonomously [100]. This real-time monitoring and repair capability closely mimic the sensory feedback and self-repair mechanisms observed in natural organisms [100,101]. By integrating sensors, these materials can detect changes in their environment and respond accordingly, enhancing their ability to detect and repair damage in prosthetic limbs.

3) The conclusions and future perspectives should encompass the primary advantages and disadvantages of the biomimetic solutions employed.

Res: As suggested, we have revised the conclusions and future perspectives section to provide a comprehensive analysis of both the benefits and limitations of biomimetic approaches in prosthetic limb development.

(Line 548)

In conclusion, biomimetics is guiding revolutionary advancements in prosthetic limb development, ushering in an era where bio-inspired prosthetics closely emulate the intricate form and function of natural limbs. By strategically employing biomimetic design strategies and integrating with biological systems, researchers and prosthetists are reshaping prosthetic technology, profoundly impacting the lives of individuals with limb loss or impairment. The profound insights gained from studying biological systems, including musculoskeletal structures, sensory feedback mechanisms, and control systems, have paved the way for groundbreaking prosthetic designs prioritizing naturalistic movement and seamless interaction. Integration with biological systems, such as osseointegration and neural interfaces, marks a new epoch of prosthetic functionality, offering users enhanced stability, control, and sensory feedback.

Moreover, the pursuit of customization and personalization has provided prosthetic users with tailored solutions optimizing comfort, functionality, and aesthetic appeal. Leveraging technologies like 3D scanning, computer-aided design, and additive manufacturing, prosthetists create devices that harmonize seamlessly with the user's anatomy and lifestyle, fostering heightened satisfaction and quality of life. Looking ahead, the horizon of biomimetic prosthetics holds boundless promise, propelled by ongoing research and interdisciplinary collaboration. Continued advancements in materials science, robotics, neural interfaces, and sensor technologies will unveil prosthetic devices with unprecedented functionality, realism, and integration. Interdisciplinary partnerships between researchers, clinicians, engineers, and prosthetic users will nurture a holistic approach to prosthetic development, ensuring that future innovations cater adeptly to diverse user needs and preferences. In summary, biomimetics is driving profound metamorphosis in prosthetic limb development, aiming to replicate the elegance and efficiency of natural biological systems. As the field evolves, bio-inspired prosthetics will play a pivotal role in augmenting mobility, fostering independence, and enriching the quality of life for individuals with limb loss or impairment. This vision foresees a future where prosthetic technology transcends restoration to enable seamless integration with the human body.

4) I think it would be good to include in this prayc a diagram showing the path from laboratory work to industrial/commercial scale transition regarding biomimetic prosthesis applications. Here it would also be useful to refer to the most recent patent literature.

Res: Thank you for your suggestion. While including a diagram illustrating the path from laboratory work to industrial/commercial scale transition for biomimetic prosthesis applications and referencing recent patent literature could be valuable, in this particular manuscript, we have opted to focus primarily on the scientific and technical aspects of biomimetic prosthetic limb development. Given the complexity of this topic and the depth of scientific content covered, we believe that providing detailed descriptions and analyses of the research and technological advancements in the field would offer readers a comprehensive understanding of the subject matter. Therefore, we have decided not to include a diagram in this instance. However, we appreciate your input and will certainly consider incorporating such visual aids in future publications where they may enhance clarity and comprehension.

5) In Table 1, the first column from the left should be labeled with a name, such as 'Type of Prosthesis', as it does not currently indicate a trend.

Res: As suggested, we have revised the first column of Table 1.

6) An important aspect in developing the use of an innovative prosthesis is to study, among other things, the cytotoxicity of its components when in contact with the body's cells. Therefore, it would be beneficial to address this issue, which can be supported by the publication mentioned here: e.g. "Pranczk, J., Jacewicz, D., Wyrzykowski, D., & Chmurzynski, L. (2014). Platinum (II) and palladium (II) complex compounds as anti-cancer drugs. Methods of cytotoxicity determination. Current Pharmaceutical Analysis, 10(1), 2-9."

Res: Thank you for highlighting the importance of studying the cytotoxicity of components in innovative prosthetic development. We agreed that addressing this issue was crucial, and we appreciated the reference provided. We incorporated relevant information on cytotoxicity determination methods and considerations into our manuscript to ensure a comprehensive discussion on the safety and biocompatibility aspects of biomimetic prosthetic materials. (line 537 and ref. 102)

7) Minor editing of English language required.

Res: Thank you for the feedback regarding the English language in our manuscript. We carefully reviewed and addressed any minor editing issues to ensure clarity and coherence throughout the text. And it was additionally reviewed through English corrections by a native-speaking colleague.
